

# Randomly correcting model errors in the ARPEGE-Climate v6.1 component of CNRM-CM: applications for seasonal forecasts

Lauriane Batté[1] and Michel Déqué[1]

[1]CNRM-GAME, Météo-France/CNRS

*Correspondence to:* Lauriane Batté (lauriane.batte@meteo.fr)

**Abstract.**

Stochastic methods are increasingly used in global coupled model climate forecasting systems to account for model uncertainties. In this paper, we describe in more detail the stochastic dynamics technique introduced by Batté and Déqué (2012) in the ARPEGE-Climate atmospheric model. We present new results with an updated version of CNRM-CM using ARPEGE-Climate v6.1, and show that the technique can be used both as a means of analysing model error statistics and accounting for model inadequacies in a seasonal forecasting framework.

The perturbations are designed as corrections of model initial tendency errors estimated from a preliminary nudged re-forecast run over an extended reference period of 34 boreal winter seasons. Perturbations are then drawn randomly in forecast mode, but consistently for all three prognostic variables perturbed. Statistical analysis of these model corrections show that they are mainly made of intra-month variance, justifying the use of these corrections as in-run perturbations of the model in seasonal forecasts. However, the inter-annual and systematic error correction terms cannot be neglected. We explore therefore the impact of using monthly mean perturbations throughout a given forecast month in a first ensemble re-forecast SMM. Time correlation of the errors is limited, but some consistency is found between the errors of two or three consecutive days. This leads us to explore the use of five-day sequences of perturbations in a second ensemble re-forecast S5D. Both experiments are compared in the light of a REF reference ensemble with initial perturbations only.

A comprehensive forecast quality analysis is then provided. Results are contrasted depending on the region and variable of interest, but very few areas exhibit a clear degradation of forecasting skill with the introduction of stochastic dynamics. We highlight some positive impacts of the method, mainly on Northern Hemisphere extra-tropics. The 500 hPa geopotential height bias is reduced, and improvements seem to project onto the representation of North Atlantic weather regimes in S5D. A modest impact on ensemble spread is found over most regions, which suggests that this method could be complemented by other stochastic perturbation techniques in seasonal forecasting mode.

## 1 Introduction

Handling uncertainties in seasonal predictions with numerical models is an issue of the utmost importance. These uncertainties arise from two main sources: initial conditions of the different variables describing the evolution of the atmosphere, ocean, and land surface, and approximations made in the modelling process. The first source is addressed by using ensemble predictions,





to sample the error on the initial state by running several integrations of a given season. The second source is now increasingly tackled in coupled global circulation models (GCMs) with several approaches developed over the last decades. Multi-model forecasts are now issued routinely by the EUROSIP consortium (Vitart et al., 2007), the United States National Multi-Model Ensemble (Kirtman et al., 2013) or the APEC Climate Center (Wang et al., 2009). Pooling several models together provides

a first rough estimate of the uncertainties related to choices in parameterizations of sub-grid processes or numerical approximations in the individual models (e.g. discretization in time and space). Numerous studies in the framework of international research projects based on retrospective seasonal forecasts (or "re-forecasts") have illustrated the gain in terms of forecast skill when using a multi-model ensemble versus a single model (see Hagedorn et al., 2005; Doblas-Reyes et al., 2009; Alessandri et al., 2011; Batté and Déqué, 2011). Further calibration of these forecasts (by weighting each individual model contribution

using a separate training period) can improve this effect (Rodrigues et al., 2013; Doblas-Reyes et al., 2005).

Simultaneously to these multi-model studies, other techniques to account for model inaccuracies were developed in the climate modelling framework. Multi-parameter (Collins et al., 2006) or multi-physics techniques (Watanabe et al., 2012) generate ensemble simulations with different physics parameter settings and physics schemes for the sub-grid scales, respectively. Over the last twenty years, stochastic perturbations have also been tested as a means of introducing noise in numerical weather

prediction (NWP) models and components of GCMs. Most studies have focused on the atmospheric component, building on methods perturbing parameterization tendencies (Buizza et al., 1999) or scattering kinetic energy dissipated by the model at the sub-grid scale back to larger scales (Shutts, 2005).

Stochastic perturbations in the atmosphere have been shown to improve the skill, reliability and mean state of seasonal forecasting systems (see e.g. Weisheimer et al., 2011, 2014; Berner et al., 2008; Batté and Doblas-Reyes, 2015). An increasing

number of studies report results from introducing stochastic perturbations in the other components of the climate system, such as the ocean (Brankart, 2013; Brankart et al., 2015), land-surface (MacLeod et al., 2015) or sea ice models (Juricke et al., 2013). Berner et al. (2015) provides a review of some of the latest advances in stochastic parameterization for NWP and climate models.

At CNRM-GAME, an alternative method to the stochastic physics techniques was designed to perturb the atmospheric

component of the coupled climate model in a seasonal forecasting framework (Batté and Déqué, 2012). Past studies (Yang and Anderson, 2000; Barreiro and Chang, 2004; Guldberg et al., 2005) had suggested that systematically correcting model tendency errors in GCMs could impact the model mean state and in some cases improve the model prediction skill. D'Andrea and Vautard (2000) had showed in a quasi-geostrophic model framework that correcting in-run flow-dependent model errors based on flow analogues could improve the model mean state. In this method, dubbed "stochastic dynamics", we apply additive perturbations

to the prognostic variables of the model drawn from a sample of model error corrections estimated in a preliminary run, instead of a systematic correction. In Batté and Déqué (2012), we showed a reduction of systematic error in the extra-tropical geopotential height fields for boreal winter re-forecasts over an extended period with CNRM-CM5. Since then, the method has been more thoroughly assessed in subsequent versions of the coupled model in a seasonal re-forecasting framework. Different choices in the frequency and strength of perturbations have been extensively tested. Building on the conclusions from these



assessments and operational constraints, a version of stochastic dynamics was introduced in the operational seasonal forecasting system 5 at Météo-France in 2015.

The aim of the present paper is twofold: first of all, illustrate that the stochastic dynamics technique can be used as a means of estimating and assessing model error. We then wish to provide a more thorough analysis of the technique in a seasonal forecasting framework with a more recent version of the coupled climate model CNRM-CM and two possible choices of perturbation frequencies and sampling.

Section 2 describes the CNRM-CM model setup for seasonal re-forecasts and provides more details on the stochastic dynamics technique. A statistical analysis of the model errors estimated from the nudged re-forecast run is led in section 3. Section 4 examines the impact of using corrections of these model errors in two stochastic dynamics seasonal re-forecasts, using a reference unperturbed run as a benchmark. Common skill and forecast quality metrics will be used, as well as an analysis of the representation of North Atlantic weather regimes. Section 5 summarizes conclusions and discusses limitations and future plans for stochastic perturbations in CNRM-CM.

## 2 Model and methods

### 2.1 CNRM-CM

The CNRM-CM global coupled model used in this study is derived from the CMIP5 version described by Voldoire et al. (2013). The ARPEGE-Climate atmosphere component is version 6.1.0, which benefits from a new prognostic convection scheme (PCMT; Piriou et al. (2007) and Guérémy (2011)), ozone and quasi-biennal oscillation parameterizations (Cariolle and Déqué, 1986; Lott and Guez, 2013) and an increased vertical resolution of 91 levels. The ocean model is NEMO version 3.2 (Madec, 2008) as in CNRM-CM5. Land surface is modelled with the ISBA-3L land surface model (Noilhan and Mahfouf, 1996) included in the SURFEX v7.3 surface modelling platform, and the sea ice component is an updated version of the GELATO sea ice model (Salas y Melia, 2002).

In this study, several hindcasts were run starting from November 1st 1979 to 2013. Initial conditions are provided by the ERA-Interim reanalysis for the atmosphere (Dee et al., 2011), ORA-S4 ocean reanalysis for the ocean (Balmaseda et al., 2013), and outputs of a coupled model run nudged towards ERA-Interim in the atmosphere and ORA-S4 in the ocean to initialize the sea ice and land surface components.

### 2.2 Stochastic dynamics

The stochastic dynamics method was first described in Batté and Déqué (2012). The idea behind this method is to combine an ad-hoc correction technique with the introduction of in-run random perturbations in the atmospheric model. It is impossible to know ahead of time the errors the model will make at each time step, however, the statistical properties of model errors can be inferred, provided we have a sufficient sample of past forecasts. Model error corrections can then be drawn at random in forecast mode. In this method, the estimation of model tendency error corrections relies on newtonian relaxation (or nudging) as





in Guldberg et al. (2005). Random model perturbations are then drawn from a population of initial tendency error corrections and applied in-run to ARPEGE-Climate. The perturbed variables are ARPEGE prognostic variables temperature, specific humidity and vorticity.

We chose not to perturb the rotational component of winds to let the model adjust to pertubations, as suggested by Guldberg et al. (2005). Another prognostic variable we did not nudge was sea-level pressure, since our philosophy was to let the surface free of perturbations so it could adjust to the higher levels in the atmosphere. Nudging of these two additional variables was tested with another version of the model, and very little difference was found in terms of model skill in seasonal re-forecast runs using the perturbations for all prognostic fields.

Equation 1 describes the nudging technique as implemented in ARPEGE-Climate, where $X$ is the vector of model prognostic variables, $\mathbf{M}$ the atmospheric model operator, and $\tau$ the relaxation time.

$$\frac{\partial X}{\partial t}(t) = \mathbf{M}(X(t),t) + \frac{X^{\mathrm{ref}}(t) - X(t)}{\tau} \tag{1}$$

In this study the prognostic fields $T$, $q$ and $\Psi$ are weakly constrained towards reference ERA-Interim data: $\tau$ is set to thirty days for each field. The rationale behind this is to let the model adjust and avoid spin-up problems due to differences between the model climate and ERA-Interim, although the drawback is a slight loss of accuracy on the tendency estimates for the model. A too strong relaxation would force the model to stay too close to the reanalysis data and far from its own attractors in forecast mode. Granted that $\tau$ is quite large, the same value was chosen for all three prognostic fields. As in Batté and Déqué (2012), the relaxation coefficients are progressively tuned down to zero in the lower levels of the model to avoid shocks at the coupling interface.

Nudging is applied during a preliminary one-member seasonal run for November to February (NDJF), starting each year from 1979 to 2012. This run serves primarily one purpose: providing the model tendency error estimates that then make up the population of random corrections from which perturbations can be drawn. Correction estimates are defined each day following equation 2.

$$\delta X(t) = \frac{X^{\mathrm{ref}}(t) - X(t)}{\tau} \tag{2}$$

The in-run perturbations in the actual seasonal re-forecasts are applied by drawing a random date $\tilde{t}$ and adding the corresponding tendency error corrections to the standard model formulation (following equation 3).

$$\frac{\partial X}{\partial t}(t) = \mathbf{M}(X(t),t) + \delta X(\tilde{t}) \tag{3}$$

Note that in a retrospective forecast framework, one could theoretically draw the correction corresponding to the time for which the model is integrated. Although one would need to draw all the consecutive corrections for the model to follow closely the reference data, corrections for a given month and year have an inter-annual component, and Batté and Déqué (2012) showed



that drawing corrections from within the year one is trying to forecast gave significantly higher skill scores. To avoid over-estimating model skill, since the re-forecast and nudged run periods are the same, the technique is applied in cross-validation mode in the re-forecasts discussed in part 4, by systematically discarding the corrections for the year being forecast from the perturbation population. Ideally, the corrections should be computed over a completely separate period from the re-forecasts.

However, when evaluating seasonal forecasting systems, a limited number of data points is available in the verification scores and we chose to use an extended re-forecast period to ensure as much robustness in our skill assessments as possible.

## 3 Analysis of ARPEGE-Climate model errors

The technique described in this study can be used as both a diagnosis of model errors and a perturbation method. The first opportunity is explored by deriving standard statistics of the ARPEGE-Climate model errors in a coupled initialized prediction

framework.

### 3.1 Spectral analysis

The $\delta X$ population is originally in spectral space (for a total wavenumber of 127) and was first analyzed in terms of squared amplitude for each total wavenumber $n$. For each prognostic variable, model level $z$ and re-forecast month $mo$ we compute $A_n(z, mo)$:

$$A_n(z, mo) = \sum_{m=-n}^{n} \left[ \frac{1}{N} \sum_{i=1}^{N} \delta X_i(n, m, z, mo) \right]^2 \quad (4)$$

where $N$ is the size of the perturbation population $\{\delta X_i\}$ for month $mo$, and $m$ is the zonal wavenumber.

To present information in a synthetic way, these statistics are integrated over 200 hPa deep layers of the model. We take into account the influence of lead time on results, since the weak nudging may allow the model to drift slowly from its initial state. Figure 1 shows results for all three nudged prognostic variables. Amplitude is plotted against the wavenumber on a logarithmic

scale for both axes. The first row shows the amplitude spectra of $\delta X$ for January corrections integrated over 200 hPa layers. For humidity (Fig. 1(a)), corrections have (as expected) an amplitude that is several orders of magnitude smaller for the upper layers of the atmosphere than for the lower layers. This difference in amplitude is much less pronounced for temperature and streamfunction. For temperature (Fig. 1(b)), it is worth mentioning that the slope of decrease in amplitude with wavenumber in log-log space is more pronounced for the upper layers of the atmosphere than for the lower layers. In the lower layers, the land-

sea contrast in temperature corrections generates small structures in the perturbation patterns, increasing the amplitude of the corrections for the higher wavenumbers. Figures 1(d–f) show the month-by-month results for the mid-troposphere layer (600-800 hPa). For all three variables, the amplitude of corrections seems to increase with lead time for the smaller wavenumbers, but a clear difference is found mainly between November and the following months of the nudged re-forecasts used to derive the correction terms.



## 3.2 Gridpoint analysis

The spectral $\delta X$ fields were then converted to gridpoint space for a spatial analysis of the correction terms. Again, results are integrated over 200 hPa layers for the sake of clarity. Figure 2 plots the December mean (in color) and standard deviation (isolines) for $\delta X$ specific humidity, temperature and streamfunction corrections for these layers.

5     As shown before, corrections for humidity are several orders of magnitude higher for the lower levels of the atmosphere than in the stratosphere, whereas temperature and streamfunction corrections are of similar amplitude. Results are consistent with the spectral analysis in Fig. 1, in the sense that for streamfunction corrections are somewhat larger in the upper layer of the atmosphere, but with less small-scale patterns, therefore concentrated on the smaller wavenumbers.

    In terms of standard deviation, patterns for temperature and streamfunction are mainly zonal (with some exceptions due to 10   land-sea contrast in the lower layers for temperature). Standard deviation increases with latitude in the northern and southern hemispheres for both variables, and values are quite similar between layers. For specific humidity, standard deviation is higher in the tropics and around the Equator. Less zonal symmetry is found than for temperature corrections. For temperature, standard deviation values are of the same order of magnitude as the mean corrections in the tropics, whereas streamfunction and humidity correction standard deviations are higher than the mean correction in most areas of the globe. The temperature mean correction 15   is mostly negative, implying that the model is warmer than ERA-Interim over most of the atmospheric column.

## 3.3 Temporal analysis

A question we wish to address when studying the perturbation population used in our forecasts is the consistency in time of the $\delta X$ terms. Indeed one possibility in the use of the perturbations is to apply corrections estimated for consecutive days in the nudged run. This would make sense only if some coherence in time is found between the $\delta X$ terms. We estimate this by com-20   puting the autocorrelation of correction terms according to the lag between their corresponding dates in the nudged re-forecast run. Figure 3 shows autocorrelation at lags of 1, 2 and 3 days of February specific humidity corrections (at approximately 850 hPa) and streamfunction corrections (circa 500 hPa), computed for all years of the re-forecast period.

    Autocorrelation for humidity corrections is generally stronger over land than ocean, and strongly decreases between one and two day lags. Some areas of the globe such as the Southern Ocean exhibit no autocorrelation even at day one. Temperature 25   corrections (center column of Fig. 3) show higher autocorrelation than humidity corrections for each time lag. The geographical areas of high autocorrelation at approximately 850 hPa are generally consistent with those of humidity corrections.

    For streamfunction, autocorrelation from one day to the next is higher than for humidity and temperature (over 0.6 in most parts of the globe), and remains above 0.4 in some areas for a two day lag. Values are typically the same order as that of humidity with a difference in the lag of one day. This shows that mid-troposphere streamfunction corrections exhibit more 30   consistency in time than lower troposphere humidity or temperature. The autocorrelation in the streamfunction correction is a motive for testing consecutive corrections over the time span of synoptic weather regimes for instance. In this paper we chose to test five day consecutive corrections as will be discussed in the next section.



### 3.4 Variance decomposition

When using pseudo-random correction terms as perturbations in an ensemble forecasting framework, we wish to combine two effects: correction of systematic errors the model makes in coupled seasonal forecasting mode, and introduction of perturbations to account for the model uncertainties that cannot be dealt with deterministic methods. Both effects could in some sense cancel

each other out: the introduction of too large purely random terms can move the model too far from its own equilibrium and induce adverse effects, which could translate into increased systematic errors in climate forecasts. On the other hand, if the systematic error correction is too strong with respect to the purely random part of the perturbations added in the model, ensemble members will follow too similar trajectories drawn towards the reference climate. In the following paragraph, we take a deeper look at the perturbations in terms of variance and mean, so as to estimate the relative importance of the systematic

error term and the interannual and intra-month (more random) variance terms in the corrections used.

Equations 5–7 show how the mean square correction terms for a given month (lead) of the nudged re-forecast can be split into three components: one is the squared mean correction, the other two the straightforward variance decomposition into inter-annual and intra-month variance. In these equations, $N$ is the total number of perturbations for a given forecast time (month), $y$ a given year of the re-forecast period used in the nudged run and $n_y$ the number of perturbations for the month

of focus in year $y$ (not the same each year in the case of February). The squared mean term $\overline{\delta X}^2$ can be interpreted as the systematic error correction for the variable studied. The variance decomposition separates the inter-annual signal (which is, to some extent, what one wants to predict with seasonal forecasts) from intra-month variability which can be approximated as noise on a seasonal time scale.

$$\overline{\delta X^2} = \frac{1}{N}\sum_{i=1}^{N}\delta X_i^2 = \overline{\delta X}^2 + \mathrm{Var}(\delta X) \tag{5}$$

$$\mathrm{Var}(\delta X) = \frac{1}{N}\sum_{i=1}^{N}\left(\delta X_i - \overline{\delta X}\right)^2 = \frac{1}{N}\sum_{y}\sum_{i_y=1}^{n_y}\left(\delta X_{i_y}^{(y)} - \overline{\delta X}^{(y)}\right)^2 + \sum_{y}\frac{n_y}{N}\left(\overline{\delta X}^{(y)} - \overline{\delta X}\right)^2 \tag{6}$$

$$\overline{\delta X^2} = \overline{\delta X}^2 + \mathrm{Var}_{\mathrm{inter}(y)}(\delta X) + \mathrm{Var}_{\mathrm{intra}(y)}(\delta X) \tag{7}$$

Figure 4 plots the relative importance of each term in the decomposition, zonally averaged and integrated over 200 hPa deep layers. The intra-month variance (blue line) is the most important component of the correction term decomposition for all layers and latitudes, except for near-surface southern subpolar latitudes in the case of specific humidity and southern polar areas in the

case of stratospheric streamfunction. In most areas, for all three variables, the intra-month term accounts for more than 50% of the total squared correction. Red lines show the proportion of inter-annual variance in the decomposition, which stays below 40% for all latitudes and layers. Although this term is smaller than the intra-month "noise", it contains valuable information for seasonal forecasts: this was shown in Batté and Déqué (2012) with a so-called "OPT" experiment where corrections were drawn in the current season of the reforecast. The black line shows the proportion of the systematic correction in the total

squared correction term. This term ranges on average between 10 and 30% depending on the variable and vertical layer. More zonal variability is found than for the inter-annual term, and the symmetry with the intra-month term is quite striking.





This analysis shows that the corrections used are mostly made of noise (at least at a seasonal time scale), although mean corrections and inter-annual variability cannot be neglected. These conclusions justify the use of these corrections as possible "pseudo-stochastic" perturbations to the ARPEGE-Climate atmospheric model in seasonal integrations.

## 4   Impact of perturbations on CNRM-CM seasonal re-forecasts

The potential of the technique is evaluated in an updated version of CNRM-CM5 for seasonal forecasts over a 34-year hindcast period.

### 4.1   Experimental setting

To evaluate the impact of this perturbation method, several sets of seasonal re-forecasts were run, starting on November 1st 1979 to 2012 and running for four months (until end of February). Re-forecast ensemble size is set to 30 members. Table 1
summarizes the characteristics of each ensemble.

Unlike Batté and Déqué (2012), where perturbations were drawn at daily intervals, we chose to run an ensemble using perturbations from 5 consecutive days, drawn separately for each member from within the other years of the re-forecast period. This experiment is called S5D. Every five days, another five day set of $\delta X$ terms is picked for each member from the same calendar month as the re-forecast. Note that the $\delta X$ terms are drawn according to the date of the nudged re-forecast run,
meaning that perturbations for the three prognostic fields are consistent with a certain model error at a given date and time.

Given the relative importance of systematic error and interannual variance with respect to total squared mean perturbations (Fig. 4), we also chose to test the impact of perturbing without intra-month variance in the corrections used. To do this we ran experiment SMM, where monthly means of $\delta X$ terms from the same calendar month but other years of the re-forecast period are used for each ensemble member. The year from which perturbations are drawn changes each month of the re-forecast.

### 4.2   Mean state

One key aspect we wish to assess when introducing such a method in a coupled model forecasting framework is how it affects the mean state of the model. Given the nature of perturbations, the impact on ensemble spread will also be considered. Although results from section 3.4 suggest that perturbations are made up mostly of intra-month variance, with a systematic error correction term accounting for less than 20% of the squared corrections in most cases, atmospheric models are highly
non-linear, and including these perturbation terms could have adverse effects.

The top row of Fig. 5 shows the mean bias for DJF sea surface temperature (left) and total precipitation (right) re-forecasts in the REF ensemble. (For areas with sea ice the model SST field is in fact the ice surface temperature, hence the large negative bias with ERA-Interim reference data.) The CNRM-CM re-forecasts exhibit typical warm SST biases along the eastern parts of ocean basins, as in the Gulf of Guinea and in the Niño 1 and 2 areas. The model also exhibits warm biases over the Southern
Ocean and along the Gulf Stream. Figures 5 (c) and (e) show the sea-surface temperature relative absolute bias for experiments SMM and S5D, respectively. Blue (red) areas indicate where bias is reduced (increased) in amplitude, regardless of the sign





of the bias of REF re-forecasts. Both stochastic dynamics methods exhibit strinkingly similar effects on SST bias: bias is increased over most of the tropical southern hemisphere ocean basins and decreased over most of the Northern Hemisphere oceans. The bias is also decreased over the Equatorial Central Pacific. Elsewhere, such as over the Southern Ocean, very little impact is found.

For precipitation, results in terms of relative bias are quite similar for experiments SMM (Fig. 5 (d)) and S5D (Fig. 5 (f)). Both versions of stochastic dynamics seem to have very little impact or slightly decrease precipitation biases (although mainly over oceans), with the exception of the Sahel and Arctic regions where the bias increases, as well as over areas of the Central and Eastern Tropical Pacific.

Supplementary Fig. S1 shows the REF biases and SMM and S5D relative biases for the first month of the re-forecast. SST
biases are already present but develop mainly after the first month of the forecast, whereas precipitation biases are already as strong in November as for longer lead times. In terms of relative bias, the stochastic dynamics technique amplifies SST biases in November in most regions of the Tropics, and seems to have a positive effect on precipitation biases already in the first month of the re-forecast.

Results for 500 hPa geopotential height are shown in supplementary figures S1 for November and S2 for DJF. Except for
parts of Eurasia, where biases (which were quite limited in REF) are amplified with both stochastic dynamics methods due to a shift of the bias pattern, both SMM and S5D exhibit lower Z500 biases than REF. Figure 6 shows the Z500 bias in experiments REF, SMM and S5D over the Northern Hemisphere extra-tropics. This figure can be compared to figure 1 in Batté and Déqué (2012). With CNRM-CM5.2, DJF Z500 bias was quite different to the bias found in REF with a more recent version of the ARPEGE-Climate model. The model now exhibits a bias quite similar to the North Atlantic Oscillation pattern, and a positive
bias over the Arctic regions where the bias was previously negative. However, regardless of this change in sign of the bias, the stochastic dynamics technique reduces the model bias over the Northern Hemisphere. Results with the new version of the model suggest that improvements in the representation of North Atlantic atmospheric circulation could be found. This aspect will be discussed later on in this manuscript.

### 4.3   Spread and deterministic skill

Ensemble seasonal forecasts with GCMs are often overconfident in the sense that the spread around the ensemble mean is smaller than the root mean square error of the ensemble mean with respect to verification data (Shi et al., 2015). This lack of dispersion in ensemble forecasts can incur misleading unreliable forecasts (Weisheimer and Palmer, 2014). Including stochastic perturbations in the components of the GCM can help partly correct these flaws, as they tend to increase the ensemble spread. In this paragraph, we wish to assess how the stochastic dynamics technique impacts ensemble spread, in the sense that this
technique is not a random perturbation technique, but rather includes model corrections. An increase in spread with the use of this technique is not straightforward, although we have shown previously that the variance of the perturbations is mainly composed of intra-month variance which we assume has a similar effect than adding noise to the system.

Figure 7 shows the ensemble spread (computed as the standard deviation around the ensemble mean) for DJF near-surface air temperature, precipitation and Z500 in experiment REF as well as the relative spread for these variables in experiments



SMM and S5D. Results in terms of the impact of stochastic dynamics on spread depend very little on the frequency and use of sequences of perturbations, as both experiments SMM and S5D yield similar results for all three variables studied in terms of geographical distribution of impacts. Spread for the SMM experiment is generally slightly higher than for S5D.

For near-surface temperature, the REF ensemble spread is large over the Northern Hemisphere extratropics in winter. This could be due to inconsistencies in the surface initial conditions with the version of the surface model used in this version of the coupled model, but this is beyond the scope of this paper. Spread is increased almost everywhere with the introduction of stochastic dynamics, except over parts of Europe, North America and the Amazon rainforest. However, in most regions the spread with stochastic dynamics is not significantly larger than without (significance at a 95% level is tested with bootstrapping intervals).

In the case of precipitation, the impact is less systematic. Regions in the Northern Hemisphere high latitudes and the Eastern Tropical Pacific exhibit a significantly higher spread with stochastic dynamics, but extended regions of North and West Africa show a lower spread in precipitation (although for these regions precipitation amounts as well as model spread are much more limited).

The highest impact on 500 hPa geopotential height (Z500) spread is found for the Northern Hemisphere extra-tropics and subpolar regions. Z500 spread is significantly higher east of Greenland with SMM perturbations. The S5D experiment exhibits similar patterns of spread increase but very few gridpoints have a significantly higher spread than REF.

These impacts on ensemble spread are limited both in terms of amplitude and geographical regions, when compared to other stochastic perturbation methods such as SPPT (see for instance figures 5 and 6 in Batté and Doblas-Reyes (2015) for impact of SPPT on global spread of SST and precipitation with the EC-Earth v3 GCM).

## 4.4 Re-forecast skill

In the previous paragraphs, we have shown that stochastic dynamics applied in a seasonal re-forecasting framework have non-negligible impacts on the forecast mean state and ensemble spread. The next step in assessing the impact of this method on forecast quality is comparing the results in terms of skill over the re-forecast period for the three experiments REF, SMM and S5D.

One common justification for the introduction of stochastic perturbations is the lack of spread of the ensemble re-forecasts with respect to skill measured as the root mean square error of the ensemble mean. We have found some (although limited) impact of the method on ensemble spread, it is therefore worthwhile checking how the spread-skill ratio evolves with the introduction of stochastic dynamics.

$$\mathrm{RMSSS}_i = 1 - \frac{\mathrm{RMSE}_i}{\mathrm{RMSE}_{\mathrm{REF}}} \tag{8}$$

The model ensemble root mean square error (RMSE) measures the distance between predicted and observed anomalies. Figure 8 shows the RMSE for REF DJF near-surface temperature, precipitation and Z500 re-forecasts. RMSE values are generally of the same order of magnitude than the ensemble spread. Supplementary figure S3 illustrates this by plotting the



spread-skill ratio for the three variables of interest in experiments REF, SMM and S5D. For near-surface temperature, RMSE is lower than spread over most oceans, but higher over many continental areas. Precipitation re-forecasts are underdispersive over most subpolar and polar regions and the Tropical Pacific, but in tropical and mid-latitudes many areas exhibit a higher RMSE than model spread. In the case of Z500, RMSE is lower than model spread over most areas of the globe, some exceptions

include North America and parts of the North Pacific and Northwest Atlantic oceans.

The second and third rows of Fig. 8 show the root mean square skill score, or RMSSS, of experiments SMM and S5D respectively. The RMSSS for experiment $i$ is computed following equation 8, where $\mathrm{RMSE_{REF}}$ is the RMSE of experiment REF. The idea of this score is to highlight areas where the model RMSE increases (negative RMSSS) or decreases (positive RMSSS) with the introduction of stochastic dynamics, by taking the REF RMSE as a reference. A positive RMSSS indicates

an improvement of the model RMSE. A perfect score would be 1, and negative values can theoretically tend to infinity. Results for near-surface temperature (left column) are quite similar between both versions of stochastic dynamics. Improvements with both versions are found over the Eastern Tropical Pacific, Northeast Canada and over the Middle East for instance. Some improvements are more pronounced in the case of S5D, as over Southeast Asia and the Horn of Africa region, but it is difficult to say which version of stochastic dynamics gives the best results. Some areas exhibit an increase in RMSE with

stochastic dynamics, such as the areas of Antarctica, the Indian Ocean east of Madagascar, and the Bering Strait area. Results for precipitation are quite patchy, although again patterns are similar for both types of stochastic dynamics. Areas of consistent improvements include West Africa, the Arabian peninsula and Central America, but in other areas such as the Eastern Tropical Pacific, the RMSE increases with the introduction of stochastic dynamics. This area is where the ensemble spread significantly increases as shown in fig. 7 (e) and (h). In this case the introduction of stochastic perturbations is detrimental to forecast quality

in terms of RMSE, but the model spread-skill ratio is only marginally affected as shown in supplementary fig. S3. It is worth mentioning that for this region, the REF ensemble is already slightly over-dispersive before introducing perturbations.

In the case of Z500, results are generally better in the S5D experiment than SMM, with the exception of the eastern coast of the USA and Australia. For S5D many areas show improvements of the model RMSE with respect to REF (which translates into a positive RMSSS).

Overall for these three variables, results show that the stochastic dynamics technique has contrasted effects on the model RMSE depending on the region of study. However, for near-surface temperature and Z500, more areas with an increased RMSSS appear. Generally speaking, the stochastic dynamics technique doesn't seem to be detrimental for model skill in terms of RMSE. Significance of the changes in RMSE is very limited (and not shown in the figures), however, provided that both S5D and SMM experiments exhibit similar RMSSS using REF as a reference, we are confident that these results are not random

noise due to a limited ensemble size and re-forecast period.

RMSE is the quadratic distance between forecast and reference observations. Depending on the amplitude of inter-annual variations of the variable of interest, the RMSE can be low although the model does not capture its interannual variability. The correlation coefficient measures to what extent the different experiments capture interannual variations of seasonal means for the variables of interest, regardless of the amplitude, giving complementary information on the model skill. Figure 9 shows

DJF correlation for near-surface temperature and precipitation in REF, and correlation differences with REF for experiments





SMM and S5D. REF exhibits high and significant correlation for near-surface temperature over most tropical regions, and over some mid-latitudinal regions such as southern Africa, eastern North America and Scandinavia. Areas with significant correlation differences (assessed following Zou (2007)) are marked by dots. Altough patterns of correlation difference with REF are similar between both stochastic dynamics experiments, both versions have different impacts on correlation when

looking only at areas of significant skill differences. S5D seems to have more satisfying results than SMM, in the sense that areas with a significant reduction of correlation skill with respect to REF are smaller or become non-significant (as in southwest China and the north Pacific), whereas some areas such as Central Eurasia, Greenland and northeast Canada, northeast Africa and the Arabian peninsula exhibit increased skill with S5D when compared to SMM.

Results for significant correlation in REF and impacts of stochastic dynamics on correlation are much more patchy in

the case of precipitation, for which little systematic impact of the method is found. As for other state-of-the-art seasonal forecasting systems, skill is much lower than for near-surface temperature. One interesting feature is a dipole of increase in DJF precipitation re-forecast skill in the Central Pacific and decrease over the Eastern Equatorial Pacific. This can be related to the improvements of the spread-skill ratio over the former region, whereas the model is already over-dispersive over the latter region where spread and model error both increase drastically with the inclusion of stochastic dynamics.

The forecast scores shown up to this point evaluate the model ensemble mean re-forecast skill. Using ensemble forecasts provides the opportunity to derive probabilistic forecasts from the ensemble members. We investigate the probabilistic skill of the different experiments in the light of two scores, namely the Brier Score and the continuous ranked probability skill score, or CRPSS. Our probability forecasts are very straightforward: the proportion of ensemble members predicting a given event is the forecast probability of the event. The Brier Score (Brier, 1950) measures the quadratic distance between forecasts

and reference data in probability space. It can be decomposed into three terms quantifying forecast reliability, resolution and uncertainty (Murphy, 1973). Reliability diagrams for Niño 3.4 region SST exceeding the second tercile (El-Niño like events) or remaining below the first tercile (La Niña like events) are represented in supplementary Fig. S4. These diagrams show the binned forecast probabilities against the relative observed frequencies corresponding to these forecasts. Ideally, points should be aligned along the diagonal to have a reliable system. The size of the dots are proportional to how frequently such probabilities

are issued. For Niño 3.4 SST, the diagrams and Brier Score decompositions show that stochastic dynamics has a very minor impact on probabilistic skill. If anything, the technique is slightly detrimental to model reliability, although differences are not significant.

Results for near-surface temperature over Europe are shown in supplementary Fig. S5. In this case, the model exhibits no skill and is (as most seasonal forecast systems) over-confident in its predictions as shown in the reliability diagrams for REF.

The stochastic dynamics experiments exhibit an improved reliability, especially in the case of warm event re-forecasts. This is however compensated in the Brier Score by slightly degraded resolution, the SMM and S5D experiments therefore do not show skill over these regions either.

Figure 10 shows the CRPSS for T2m, precipitation and Z500 for all three experiments. CRPSS is computed at each gridpoint using ERA-Interim (or GPCP for precipitation) data of the other years of the re-forecast period as a reference (climatology)

probability forecast. As for deterministic skill scores, areas of positive skill are mostly constrained to the tropics, and precipi-





tation forecasts are very poor. The region dominated by ENSO concentrates the higher skill scores in the case of near-surface temperature. Minor improvements in the area are obtained in the SMM ensemble for both temperature and precipitation. For Z500, hints of improvements are found over North America, alongside a reduction of negative CRPSS over Europe. However, in most areas, very little change is seen between the three ensembles. Note that the scores presented here were computed based on model anomalies in cross-validation mode, but without further calibration of the ensemble forecasts (as a quantile-quantile calibration technique for instance) which can improve results with respect to climatology. The results in terms of CRPSS are consistent with the minor changes in the model spread-skill ratio and low impact of the stochastic methods on model reliability and resolution in the Brier Score evaluations shown in supplementary figures S4 and S5.

The global evaluation of the stochastic dynamics technique in terms of impact on re-forecast skill is quite contrasted, with results depending on the regions of study. Furthermore, we face a recurrent issue in the seasonal to decadal prediction field, which is the limited statistical significance of differences in skill between two versions of a system. We stress however that the results presented here are computed for relatively large ensemble sizes (30 members) and a 34-year re-forecast period, giving a certain robustness to results presented here. It is also worth mentioning that most significant impacts found with the stochastic dynamics technique are found for both versions of the method discussed in this paper. This could imply that the skill improvements are mostly due to improvements in the model mean state due to the non-zero mean term in the perturbations applied in the stochastic dynamics technique.

Earlier in this paper, we found evidence that the stochastic dynamics technique improved the Z500 bias over the North Atlantic mid-latitudes and the Arctic. The technique also improves the model spread-skill ratio over Europe (see supplementary Fig. S3 for Z500). Figure 11 corroborates this: we computed the model spread and RMSE for Z500 averaged over Europe, according to the lead time, for the three ensembles. The RMSE is reduced with the stochastic dynamics technique in the first month of the re-forecast, and spread is larger than for REF in both S5D and SMM ensembles for each re-forecast lead time.

Granted that some improvements are found both in the model mean state and spread-skill ratio for Z500 over the region, we examine in the following section the impact of the technique on the representation of North Atlantic large-scale circulation, both in terms of the re-forecast skill of the North Atlantic Oscillation (NAO) and representation of the North Atlantic-Europe weather regimes.

### 4.5 North Atlantic large-scale circulation

### 4.5.1 North Atlantic Oscillation re-forecasts

In this study we compute the NAO index as the projection of the DJF Z500 anomaly for a given year on the leading EOF of 500 hPa geopotential height in ERA-Interim over the North Atlantic - Europe region defined by Hurrell et al. (2003) over the reference period (in cross-validation mode, e.g. by removing the year of interest from the 1979–2012 period). This is done both for the ERA-Interim reference index and each member of the three re-forecast ensembles. Figure 12 shows boxplots of the REF, SMM and S5D ensemble re-forecasts of the NAO index, verified against ERA-Interim. The correlation between the ensemble mean indices and the ERA-Interim index is shown in the top left corner of the figure. Correlation in REF is reasonably high



when compared to coupled prediction systems with similar resolutions over a 30-year re-forecast period (Kim et al., 2012),
and significantly above zero. The SMM ensemble exhibits a slightly lower correlation than REF, and S5D perturbations seem
to improve correlation of the NAO, but differences are not significant when assessed with a bootstrapping technique. The
stochastic dynamics technique has no impact on the ensemble spread in the NAO index re-forecasts when computed over the
entire re-forecast period.

### 4.5.2  Weather regime statistics

The impact of stochastic dynamics on sub-seasonal variability is assessed, focusing on the North Atlantic region where a
strong decrease in systematic error was found. We examine how the model represents the four main winter weather regimes
over the region, defined following Michelangeli et al. (1995) using an EOF decomposition of daily 500 hPa geopotential height
anomalies and a k-means clustering technique. The four centroids of the weather regimes are represented in supplementary
Fig. S6. Frequency of attribution to each cluster is shown in the figure.

As in other standard-resolution climate GCMs (see for instance Dawson et al. (2012)), the seasonal forecasting system dis-
cussed here fails to represent the North Atlantic weather regimes properly. Moreover, the REF re-forecast exhibits quite strong
Z500 biases over the region. We therefore project model daily 500 hPa geopotential height anomalies for each ensemble mem-
ber onto the EOFs of the ERA-Interim anomalies instead of using the model EOFs. Weather regimes are attributed following
an euclidean distance criterion. In the following, we chose a minimum weather regime duration of 3 days, all days in regimes
lasting less than this limit were classified as regime transition days. This explains the minor differences in climatological
frequencies of the ERA-Interim regimes in table 2 and Fig. S6.

Table 2 shows the frequency and mean duration of each weather regime in ERA-Interim and experiments REF and S5D.
Compared to reanalysis data, the REF ensemble underestimates the frequency of the NAO+ regime by more than $5.5\%$ and
overestimates the NAO− regime frequency by over $4\%$. The introduction of stochastic dynamics in the atmospheric model
tends to correct at least parts of these errors, as S5D statistics are generally closer to ERA-Interim than REF. This is also
the case for regime duration. The mean duration of each regime is systematically improved with S5D perturbations. In most
cases the length of the regimes is not considerably changed, apart from the Blocking regime for which stochastic dynamics
in the S5D experiment make the regime last on average 0.4 days longer. One could think that the introduction of stochastic
perturbations could cause the model to shift from one regime to another more frequently, therefore shortening the mean length
of each regime. Results in table 2 show that this is not the case, as S5D perturbations tend to increase regime duration when
the model under-estimates it.

Another aspect we wish to assess is how the stochastic dynamics technique changes the frequency of weather regime transi-
tions. Figure 13 shows the frequency of these transitions for ERA-Interim, REF and S5D. Transitions are defined as follows:
we look at the end of a given regime (which lasts three days or more) which is the following regime. Transitions can therefore
be from one regime back into the same one, under the condition that the intermediate days are a transition (less that three
days in another regime). With respect to ERA-Interim over the same period, CNRM-CM (REF) represents reasonably well the
North Atlantic weather regime transition frequencies. Some frequencies are over-estimated, as the NAO− transition to another





NAO− event (27% in REF versus 16% in ERA-Interim), and the NAO+ to Scandinavian Blocking transition (47% in REF versus 35% in ERA-Interim). For these two examples, the S5D experiment including stochastic dynamics slightly improves results. However, this is not always the case, and it is impossible to conclude as to one experiment exhibiting better weather regime transition frequencies than another.

These results for North Atlantic weather regimes show that when including perturbations to the model dynamics, the intraseasonal variability of the model stays quite consistent with reference data, and improves in some aspects such as regime frequencies. Adding noise to the model dynamics does not significantly push the model into favoring some weather regime transitions to others.

### 4.5.3   Weather regime frequency re-forecast skill

Supplementary fig. S7 represents boxplots of the ensemble re-forecasts of the four weather regime frequencies for DJF 1979–2012 in experiments REF (left) and S5D (right). No striking impact on the ensemble spread of the weather regime frequencies is found with the introduction of stochastic dynamics in CNRM-CM. Table 3 shows the correlation between the ensemble mean frequency and ERA-Interim for each weather regime (shown by red dots for each year in fig. S7). Correlation is generally quite poor for the REF ensemble, as weather regime frequencies are quite challenging to predict at a seasonal time scale due to

internal variability. However, we do notice a significant increase in the correlation coefficient for NAO− regime frequency predictions, consistent with the improvement in the NAO index re-forecasts with S5D suggested earlier. The ensemble with stochastic dynamics seems to capture some signal for the extreme winter 2009/10 (Ouzeau et al., 2011), as shown in supplementary fig. S7. For the other three regimes, no significant change is found. This encouraging result should be interpreted with caution due to the high levels of uncertainty when dealing with seasonal re-forecasts over mid-latitudes (Shi et al., 2015).

As another way of assessing weather regime forecast quality over the re-forecast period, we computed a score based on the Brier Score over the four weather regimes by comparing the actual weather regime frequency to the weather regime probability given by the ensemble forecast. This score is a distance in probability space and should be as small as possible. A corresponding (positively oriented) skill score is obtained by computing a corresponding reference distance. We chose the ERA-Interim frequency of each regime over all other years of the re-forecast period as a reference forecast. Our REF ensemble

has a skill score of -0.011, meaning that using ERA-Interim climatology over the other years of the re-forecast gives a better probability forecast than CNRM-CM of weather regime frequencies. When introducing 5-day stochastic dynamics, the skill score is positive and reaches 0.081. Again, significance of these results is quite limited, but all seem consistent and lead us to conclude that this technique improves the representation of North Atlantic variability at a seasonal time scale.

### 5   Conclusions

This study has provided details on the stochastic dynamics technique, first developed and described in Batté and Déqué (2012) and further amended in more recent versions of the CNRM-CM coupled GCM for seasonal forecasts. A version of this method





(similar to the S5D experiment discussed in this paper) has been implemented in the next operational seasonal forecasting system 5 at Météo-France.

Stochastic dynamics is based on an estimation of atmospheric model errors using nudging, and the introduction of random in-run corrections of these model errors. The statistical analysis of model errors showed that the amplitude of spectral corrections

was highest in the smaller wavenumbers, and generally increased between the first month and the following months of the nudged re-forecast run. Unlike other stochastic perturbation techniques, the perturbations in the stochastic dynamics technique present by construction a non-zero mean and variability in both space and time which is specific to each perturbed variable. Some time consistency in perturbations can be sought by using a sequence of corrections from the nudged run, as was done for experiment S5D. A decomposition of the mean squared perturbation terms showed that perturbations consisted mainly of

intra-month variance, but that inter-annual variance and systematic part of the perturbations was non-neglectable.

Beyond the analysis presented in Batté and Déqué (2012), the impact of stochastic dynamics was studied in two boreal winter seasonal re-forecast runs compared to a reference re-forecast with initial perturbations only. The SMM experiment used monthly mean correction terms drawn seperately and each month for each ensemble member, whereas the S5D experiment explored the use of five-day sequences of perturbations drawn independently every five days for each ensemble member. Results

showed a reduction of precipitation bias over most areas of the globe, as well as improvements in the model mean Z500 field over the Northern Hemisphere. The reduction of Z500 bias is consistent with results from Batté and Déqué (2012) although this previous study used an older version of the seasonal forecasting system with different biases. In terms of forecast skill, improvements are found mostly for near-surface temperature due to an overall increase in ensemble spread. For precipitation, results are patchy and some areas such as the Eastern Tropical Pacific exhibit a decrease in skill with the introduction of

stochastic dynamics.

An evaluation of the representation of variability over the North Atlantic region was then presented, looking at both NAO forecasting skill and the representation of North Atlantic weather regimes. Encouraging improvements were found in the frequency of weather regimes and some weather regime transitions, although most differences are most likely non significant. Interestingly, the introduction of stochastic dynamics does not decrease the length of weather regimes nor significantly alter

regime transition frequencies. A considerable improvement of the correlation of DJF NAO− regime frequency with ERA-Interim was also found with the S5D experiment, although no significant change was found in DJF NAO index correlation skill. Overall, the introduction of stochastic dynamics perturbations in CNRM-CM seems to benefit the representation of North Atlantic weather regimes.

Several limitations appear with this method. The perturbations rely on *a priori* estimations of model errors by atmospheric

nudging, therefore the method requires a preliminary nudged run consistent with the target season and model version, which can be computationally expensive. However, the method is quite straightforward to implement once atmospheric nudging is included in the model. Moreover, this method requires very limited tuning with respect to other stochastic perturbation techniques, since only the strength of the relaxation in the preliminary nudged run and the frequency of perturbations in forecast mode need to be adjusted.



On more theoretical grounds, the philosophy behind the stochastic dynamics technique is very *ad hoc* in the sense that it uses model error statistics to correct these in forecast mode, instead of introducing stochasticity in the physical parameterizations of the model. The additive perturbations to the model dynamics can cause imbalance in the energy and water budgets, although the impact most likely remains quite limited, as shown by the skill assessments in this study. In terms of interactions with surface and ocean components in the coupled model, the perturbations are dialed down to zero in the lowest levels of the atmosphere, but results in terms of SST biases show that these do have a systematic impact on the surface. This aspect will be further evaluated in specific case studies. However, our belief based on comprehensive skill evaluations is that the overall influence of the technique is positive at a seasonal time scale.

One motivation for introducing stochastic dynamics in the CNRM-CM climate forecasting systems was to generate ensembles in burst mode instead of lag-average initialization. This evolution of the initialization technique enables us to use the same configuration for weekly and sub-seasonal forecasts, without significantly degrading the skill of several ensemble members by starting from older initial conditions. This study showed however that the impact of the method on ensemble spread (with respect to perturbing only at forecast time 0) depended on the area and variable of interest, and was somewhat limited. The technique could be complemented by other stochastic methods to perturb the atmospheric physical tendencies, although interactions between this type of perturbations and dynamical nudging in the model should be carefully documented. Developments are currently underway to include SPPT (Palmer et al., 2009) in the ARPEGE-Climate model.

An extension of the method considered at CNRM is to introduce flow-dependency in the corrections, based on classification of the correction population depending on the state of the atmosphere, following the idea explored by D'Andrea and Vautard (2000). Preliminary studies using classification of streamfunction fields or based on the state of ENSO gave disappointing results in re-forecast skill assessments. An interesting perspective to explore this aspect is to take advantage of the long reanalysis datasets such as ERA-20C (Compo et al., 2011) and 20CR (Poli et al., 2013), however the applications in real-time coupled forecasts would be necessarily limited since these reanalyses span periods for which ocean data are unavailable.

**Author contributions**

L. Batté and M. Déqué developed the technique and designed the experiments. M. Déqué ran the simulations discussed in this paper. L. Batté performed the analysis and prepared the manuscript with contributions from M. Déqué.

**Code and data availability**

Most parts of the codes composing the CNRM-CM model discussed in this paper, including the ARPEGE-Climate v6.1 model, are not available in open source. ARPEGE-Climate code is available to registered users for research purposes only. Outputs from the seasonal re-forecasts discussed in this paper are available upon request to the authors, and some will be included in the SPECS project repository at the British Atmospheric Data Centre (BADC, http://browse.ceda.ac.uk/browse/badc/specs/data/).





*Acknowledgements.* The research leading to these results has received funding from the European Union Seventh Framework Programme (FP7/2007-2013) SPECS project (grant agreement number 308378). Re-forecasts were run on the ECMWF supercomputer.

We are indebted to developers of R libraries s2dverification and SpecsVerification used for some analyses in this paper, as well as Python Matplotlib and CDAT packages.



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





**Table 1.** Characteristics of the seasonal re-forecast experiments discussed in this paper.

| Name | Ensemble size | Initial perturbations | Stochastic Dynamics | Characteristics |
|------|---------------|----------------------|---------------------|-----------------|
| REF | 30 | random $\delta X$ | no | - |
| SMM | 30 | none | yes | monthly mean $\delta X$ terms |
| S5D | 30 | none | yes | five consecutive $\delta X$ terms |

**Table 2.** Weather regime frequencies and mean duration (in days) for ERA-Interim and experiments REF and S5D (weather regimes are defined for a duration of 3 days or more, so frequencies don't sum up to 100%).

|  | NAO+ | | Blocking | | NAO− | | Atl. Ridge | |
|--|------|--|----------|--|------|--|------------|--|
| ERA-Interim | 32.1% | 9.48 | 24.4% | 7.14 | 18.8% | 9.27 | 16.6% | 5.85 |
| REF | 26.5% | 8.28 | 23.4% | 6.56 | 24.0% | 8.90 | 16.8% | 6.41 |
| S5D | 28.0% | 8.35 | 23.8% | 6.97 | 21.9% | 9.16 | 17.1% | 6.38 |

**Table 3.** Correlation between ensemble mean DJF North Atlantic-Europe weather regime frequencies in experiments REF and S5D and ERA-Interim. Weather regimes are defined for a duration of 3 days or more.

|  | NAO+ | Blocking | NAO− | Atl. Ridge |
|--|------|----------|------|------------|
| REF | 0.21 | −0.03 | 0.25 | −0.06 |
| S5D | 0.17 | 0.00 | 0.54 | −0.01 |





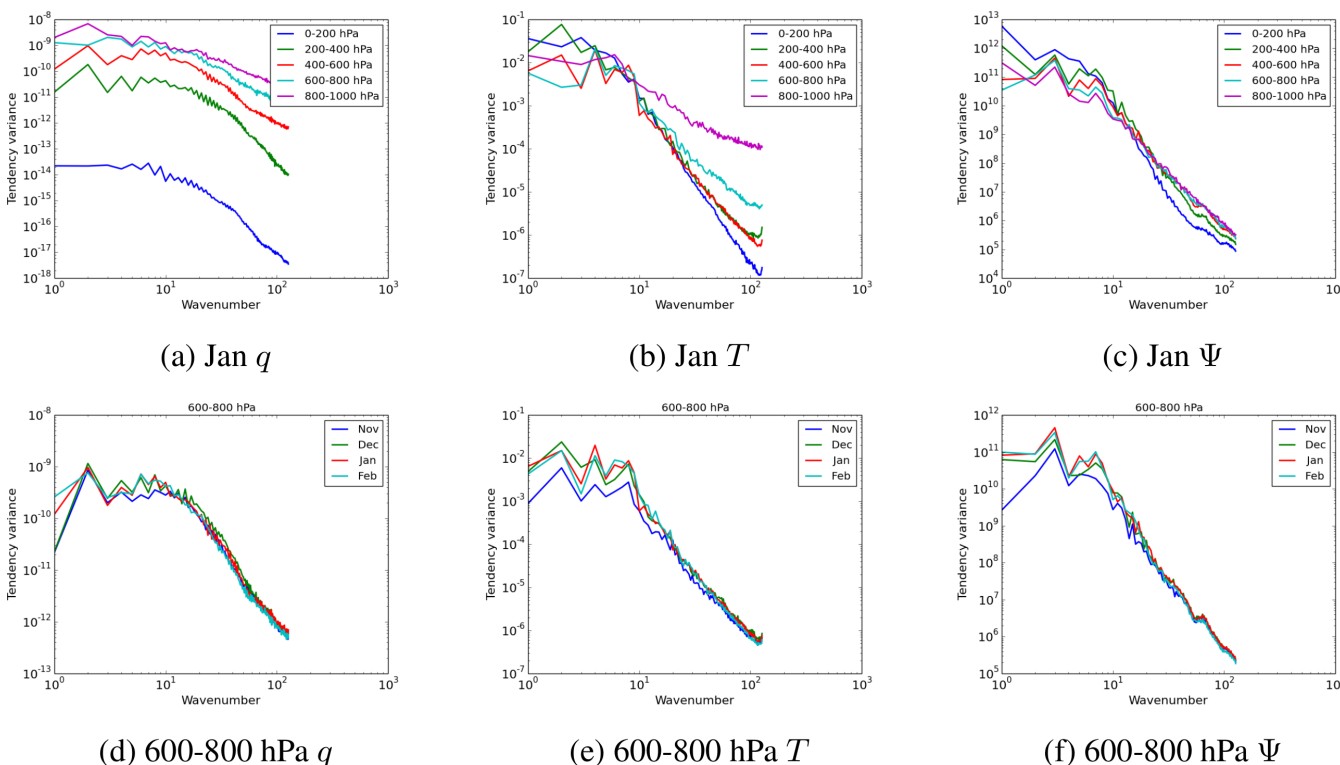

**Figure 1.** Spectral amplitude of corrections for (from left to right) specific humidity, temperature and streamfunction for January 1980–2013 integrated over 200 hPa layers of the atmospheric model (top row), and for each month of the nudged runs for the 600-800 hPa layer (bottom row). Values are calculated with raw $\delta X$ spectral fields (corrections per model time step).





$q$ · $T$ · $\Psi$

**Figure 2.** Mean and standard deviation of December 1979–2012 corrections for (from left to right) specific humidity, temperature and streamfunction for 200 hPa layers of the atmospheric model (centered from top to bottom at 100 hPa, 300 hPa, 500 hPa, 700 hPa and 900 hPa respectively). $\delta X$ values are converted to standard units per day.



**Figure 3.** Autocorrelation for lags (top to bottom) 1 to 3 days of February 850 hPa humidity (left) and temperature (center) corrections and 500 hPa streamfunction corrections (right).





**Figure 4.** Decomposition of the zonal mean square correction term for December corrections. Statistics are computed for 200 hPa layers as in Fig. 2. Black lines represent the squared mean term, red lines the interannual variance, and blue lines the intra-month variance.





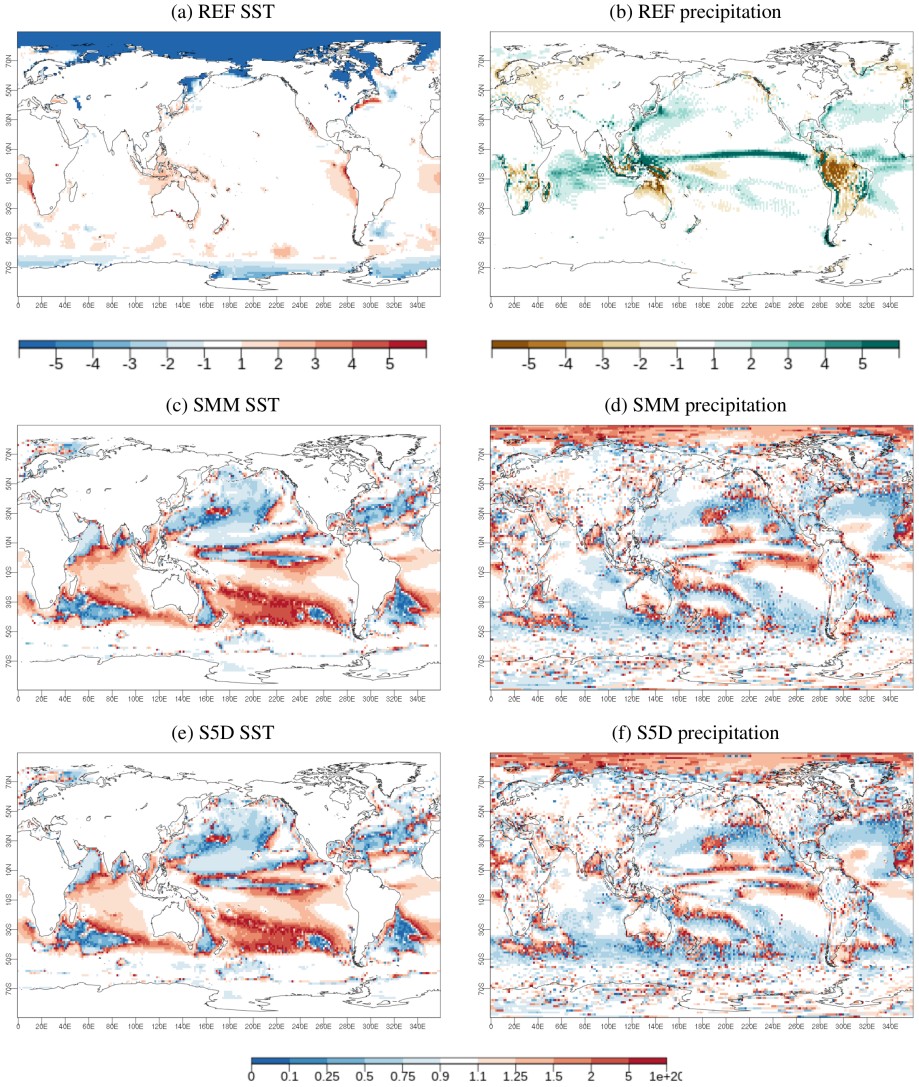

**Figure 5.** DJF bias (top row) for REF experiment SST, precipitation and Z500 (from left to right); corresponding relative absolute bias in experiments SMM and S5D (second and bottom rows, respectively). Bias is computed with respect to ERA-Interim for SST and GPCP for precipitation. Areas in blue indicate where bias is lower with respect to REF, whereas areas in shades of red show where bias is increased, regardless of the sign of the bias.





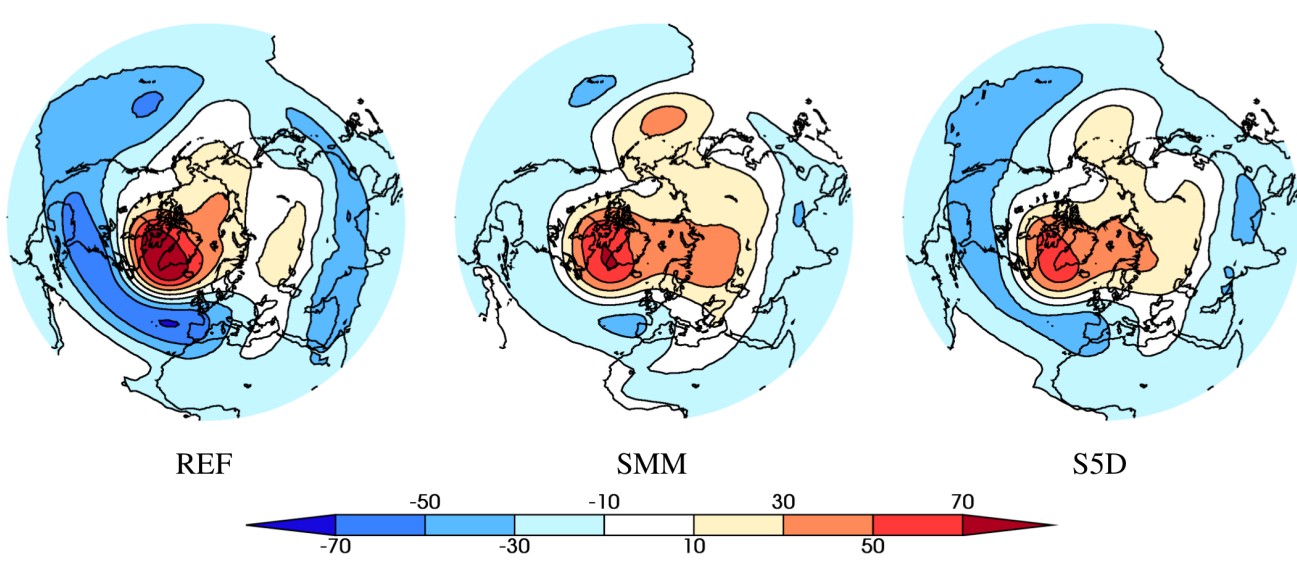

**Figure 6.** Mean bias for DJF 500 hPa geopotential height with respect to ERA-Interim (in m) over the Northern Hemisphere for experiments (from left to right) REF, SMM and S5D.





**Figure 7.** DJF spread (top row) for REF experiment near-surface air temperature, precipitation and Z500 (from left to right); corresponding relative spread in experiments SMM and S5D (second and bottom rows, respectively). Spread is computed as the standard deviation around the ensemble mean. Areas in blue indicate where spread is lower with respect to REF, whereas areas in shades of red show where spread is increased, and dots show where differences are significant at a 95% level based on bootstrapping intervals.





**Figure 8.** DJF root mean square error (RMSE) for REF (top row) computed against ERA-Interim (GPCP in the case of precipitation) over the re-forecast period for near-surface air temperature, precipitation and Z500 (from left to right). Middle and bottom rows: SMM and S5D root mean square skill score (RMSSS) using REF as a reference forecast. Areas in blue indicate where RMSE is higher than in REF, whereas areas in shades of red show where the RMSE is lower.





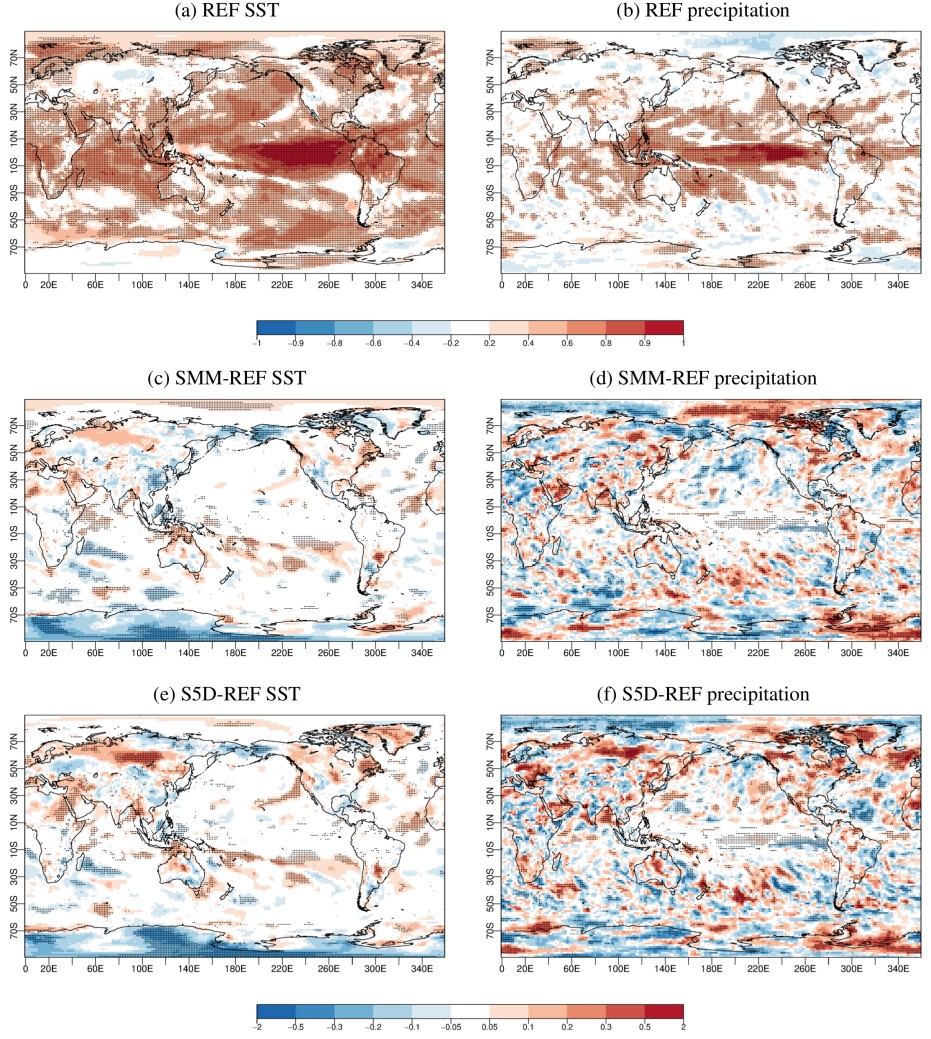

**Figure 9.** REF experiment DJF correlation (top row) for near-surface air temperature (left) and precipitation (right) with respect to ERA-Interim and GPCP, respectively. Areas with correlation significant at a $95\%$ level are marked by dots. Second (resp. bottom) row: difference in correlation between experiments SMM (resp. S5D) and REF. Significance of correlation differences (marked by dots) is assessed following Zou (2007).





**Figure 10.** DJF continuous ranked probability skill score (CRPSS) for REF, SMM and S5D experiments (top to bottom rows, respectively) near-surface air temperature, precipitation and Z500 (from left to right). Areas in red/blue indicate where the model skill is higher/lower than a reference forecast using climatology.







**Figure 11.** Evolution of spread (dots) and RMSE (lines) with forecast time for 500 hPa geopotential height over Europe in experiments REF (red), SMM (blue) and S5D (green).




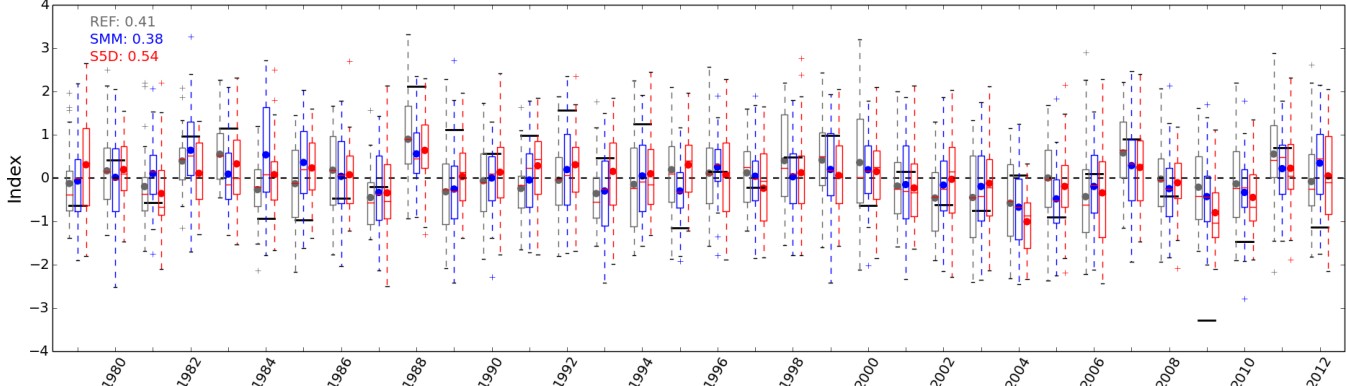

**Figure 12.** DJF NAO index computed with ERA-Interim 500 hPa geopotential height (black lines) and boxplots of ensemble re-forecasts REF (gray), SMM (blue) and S5D (red) NAO indices computed by projecting model anomalies on the ERA-Interim NAO pattern. Anomalies and NAO indices are computed in cross-validation mode. The correlation between the ensemble mean and ERA-Interim index is shown in the top left corner of the figure.

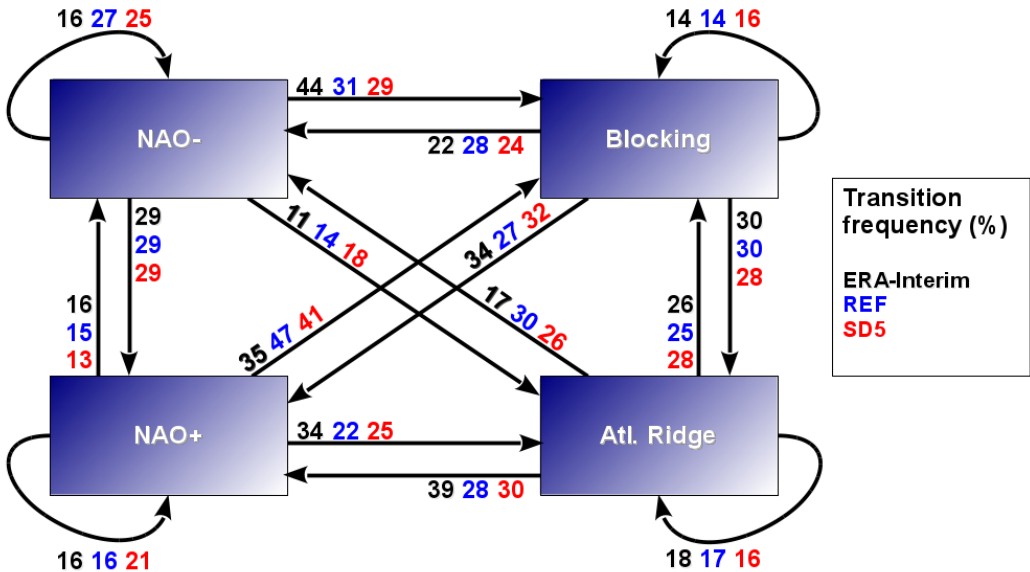

**Figure 13.** Frequency of weather regime transitions (in %) computed by discarding regimes shorter than 3 days (considered as transition days). Results are shown for ERA-Interim reanalysis and experiments REF and S5D for DJF 1979–2012.