# Peer review of "Randomly correcting model errors in the ARPEGE-Climate v6.1 component of CNRM-CM: applications for seasonal forecasts"

_Geoscientific Model Development, 2015_

## Referee Comment (RC1) · Anonymous Referee #1 · 11 Mar 2016

General comments:

This manuscript conducts a comprehensive evaluation of including a model-error representation, called stochastic dynamic technique, in seasonal ensemble forecasts. The stochastic dynamic technique randomly draws from tendencies, that were obtained as nudging tendencies by relaxing the model to reanalysis data. The valuation is among the most comprehensive evaluations I have ever seen for seasonal forecasts and I complement the authors to quantify statistical scores such as mean bias, spread, correlation as well as physical processes such as weather regimes and modes of tropical

variability. Great care was given to discuss the statistical significance of the results and the application of the tendencies and verification in cross-validation mode.

The results are described and interpreted with care. Unfortunately, the impact of the stochastic dynamic technique is small. I suggest to expand the discussion in the conclusions, why the impact is small and why the results of forcing with monthly mean tendencies is so similar to using 5d-consecutive tendencies.

Specific comments:

- Move discussion on page 13, l13-15 to conclusions and expand. Is there a pattern that SMM and S5D have similar impact on mean statistics, but S5D a larger impact on statistics involving the second moment?

- It would be interesting to see a map of a particular 5D-tendency to get a feeling for the spatial correlation scales.

- It might be helpful to plot the differences SMM-RED and S5D-REF for figures 5, 6 and 10 to see if there is a coherent regional signal. As the manuscript admits, the absolute plots look very similar.

Technical corrections:

- Figures: On many plots I could not see the dots signifying statistical significance. Maybe increasing the panel size would help? Sometimes different color schemes (saturated vs unsaturated) are used to distinguish significant regions.

- Figure 5: Caption still mentions z500 plots, which are now in the supplementary material

- p16, l10: neglectable -> negligible

- p3, l21: 'y' -> 'and'

---

## Referee Comment (RC2) · Anonymous Referee #2 · 15 Mar 2016

The manuscript applies an earlier introduced stochastic dynamics technique in a seasonal forecast system, using a comprehensive set of ensemble re-forecast simulations. These re-forecasts are assed to which extent the stochastic dynamics technique improves predictive skill and ensemble spread.

The manuscript presents novel results that are of interest to a large community. The results are presented very well, and I very much enjoyed reading the manuscript. Hence, I recommend the manuscript to be published, and I only have a few minor points which could improve the manuscript.

[Figure]

1. I commend the authors for being clear about the limitations of their technique and not overselling their results. Yet, I think the manuscript would benefit from establishing clear expectations of the technique – and the abstract, the introduction and the conclusions are not very coherent. The abstract summarizes the results mostly with respect to predictive skill. The introduction formulates (p. 3, para around line 5) poses two somewhat different aims, while the conclusions (p. 17, para around line 10) suggest some disappointment because the author's originally wanted to improve the ensemble generation.

I suggest that the authors a. formulate a coherent goal for the manuscript b. include (in addition the limitations discussion in the last section of the manuscript) a discussion where they see the further potential of the technique (based on the presented results.

Also, as a comment, I think the differences/improvements in figure 6 are not small.

2. The split up of the model (experiment) description between section 2.1 and 4.1 was not entirely intuitive to me. Could the two section be combined within section 2? Also, is the horizontal resolution mentioned anywhere? 3. I think section 4.5.1 could do with a mentioning of the recent results of NAO skill (e.g. Scaife et al., Butler et al.; including Weisheimer et al, if the authors wish to question the results). 4. This will be take care of later anyway, but I noticed that (in an otherwise very carefully wriiten manuscript) the references to figures are sometimes with 'Fig.' and sometimes with 'figure'. Also, are all supplementary figures cited (in the right order)? Maybe I overlooked it, but where is figure S2 cited?
* * *

---

## Referee Comment (RC3) · Anonymous Referee #3 · 16 Mar 2016

[a4paper]article

amsmath amsfonts

**"Randomly correcting model errors in the ARPEGE-Climate v6.1 component of CNRM-CM: applications for seasonal forecasts"**

by
Lauriane Batté and Michel Déqué
GMD - Discussions, 2016

**1 General comments**

The manuscript describes an update to and extension of the stochastic dynamics technique introduced by Batté and Déqué (2012). The application presented is seasonal ensemble forecasting of 34 boreal winter seasons. The idea is to first calculate a database of approximate initial tendency errors for temperature, vorticity and specific humidity. This is done by weakly nudging the ARPEGE-Climate v6.1 model state towards ERA-Interrim re-analyses for all winter seasons, and then letting the nudging term be the estimate of tendency errors. Once calculated the nudging term is stored in terms of monthly means and 5 day means (two different approaches tested in the manuscript). During the seasonal ensemble forecast simulations for a given year initial tendency errors are then drawn randomly as stochastic forcing (perturbation). Only tendency

errors from other years than the one in question is used for perturbation, i.e., a proper cross-validation technique is applied. The paper analyses and discuses the statistics of the model tendency errors, the model bias for the ensemble forecasts, and the quality of the forecasts as compared to a reference set of ensemble forecasts, which were not perturbed. A main issue is if an enhanced model spread can be obtained via the nudging, and, of course, if the skill of the forecast system is improved.

The main conclusions are that there is generally a weak improvement in forecast skill and model spread.

I believe there is a general problem with the use of the "initial" when $\tau$ is as long as 30 days. With such a weak nudging this term can not be said to represent initial tendency errors but rather long term secondary adjustments (that luckily seem to have some positive impact). This is of cause because, on a monthly time scale, initial forcing in terms of e.g. potential vorticicy will show up far away via Rossby wave dispersion. As an example consider the right column of Figure 2: These corrections could very well be due to "real" initial errors in the tropics. It is therefore suggested not to use the expression "initial" tendency errors. One could, e.g., call it model drift error.

The paper is well written and represents a significant amount of careful work. Although there seem to be some positive impacts from the introduction of the stochastics dynamics the results vary a lot from region to region. So, one cannot say that the technique is the wholy grale needed to improve seasonal forecast techniques. However, it is a relevant contribution that can be combined with other stochastic techniques.

The paper can be accepted when the following minor comments have been considered.

**2 Specific comments**

Page 1, line 12: Change "SMM" to "(SMM)" and "SD5" to "(SD5)".

Page 2, line 29: Change "In this method" to "In the method presented here"

Page 4, line 4: I presume you mean "not to perturb the divergent component" instead of "not to perturb the rotational component" (since vorticity represents the rotational part).

Page 5 ff: Probably not only the magnitude but also the shape of the spectra are quite dependent on $\tau$. A short discussion on this would be relevant.

Page 8, Section 4.2: It would be relevant to show - or at least discuss - the bias in the initial nudged simulations as well. Ideally the mean error of these runs should be small. But with the large value of $\tau$ one would suspect that this is not the case.

Page 9, line 32: Replace "than adding" with "to adding".

Page 10, Eq. (8): It is suggested to move this equation down to where it is introduced in the text.

Page 11, line 32: "... not capture its interannual variability". One would guess that it could also be large if the model has a bias. Any bias could be subtracted before calculating RMSE. This would probably give considerably smaller RMSE's.

Page 13, lines 17-21: Also here it could be relevant to eliminate the impact of bias.

Page 14, line 3: You could provide a quantitative estimate of the uncertainties in the correlations!

Page 14, lines 22-23: Why is there no NMM in Table 2 (and 3)?

Page 15, Section 4.5.3: I think this section can be removed. It does not add much to the findings already described.

---

## Author Comment (AC1) · 2 May 2016

Reply to interactive comment by anonymous referee #1 by Lauriane Batté

We wish to start by thanking the reviewer for his/her constructive comments on our manuscript.

1) Reply to general comments:

"Unfortunately, the impact of the stochastic dynamic technique is small. I suggest to expand the discussion in the conclusions, why the impact is small and why the results of

forcing with monthly mean tendencies is so similar to using 5d-consecutive tendencies."

Our hypothesis, based on this study and previous work on the technique, is that the main impact of our perturbations does derive from the systematic error corrections encompassed in the perturbation term. This is why on average, 5d-consecutive tendencies have the same effect on seasonal forecast quality than the monthly mean tendencies.

Regarding the limited impact in both setups on seasonal forecasting skill, this is most probably related to the weak constraint in our preliminary experiment. With a previous version of the model, other settings for the nudged preliminary run were tested, using a stronger constraint. However, our feeling was that since we were nudging towards ERA-Interim, using too strong a nudging could be a drawback, in the sense that we would be drawing the model away from its own equilibrium (and more towards that of the ECMWF model), and the terms would be less representative of long-term model errors. Were we to have a reanalysis based on the ARPEGE-Climate model, we could consider using stronger nudging and explore the impact this has then when applying the corresponding perturbations in seasonal forecast mode.

2) Reply to specific comments:

"Move discussion on page 13, l13-15 to conclusions and expand. Is there a pattern that SMM and S5D have similar impact on mean statistics, but S5D a larger impact on statistics involving the second moment?"

I have rearranged the conclusions to take into account this comment. Regarding the impact on statistics involving the second moment, our results with respect to weather regime duration, etc. suggest that differences are also small between S5D and SMM. This could be due to the fact that our nudging is quite weak and is a perspective for future work.

"It would be interesting to see a map of a particular 5D-tendency to get a feeling for the

spatial correlation scales."

I included this in the supplementary information of the article as (new) figure S1, and commented this in the article (section 3.3).

"It might be helpful to plot the differences SMM-REF and S5D-REF for figures 5, 6 and 10 to see if there is a coherent regional signal. As the manuscript admits, the absolute plots look very similar."

Figure 5 shows the relative absolute bias of SMM and S5D with respect to that of REF in the middle and bottom rows (meaning that blue areas show where bias is reduced, and red areas where bias is enhanced, regardless of sign). Over the Northern Hemisphere extra-tropics, the main impression is that SST bias is reduced with our technique, whereas results are more contrasted for precipitation (patchy areas, general reduction of bias over the mid-latitudes, and increase in precipitation bias over the Arctic).

In my opinion, the different figures in figure 6 are not that similar, they show a substantial reduction of the Z500 bias over most of the Northern Hemisphere extra-tropics. As additional information, the figure 1 included in this comment shows the relative absolute bias for DJF Z500 over the re-forecast period for experiments SMM and S5D with respect to experiment REF. Similar information is available for the reader in the supplementary figure S2. I clarified the sentence referring to these results and figure S2 in the revised version of the manuscript.

Figure 10 now shows the CRPSS for REF with respect to reference data climatological probabilities, and CRPSS for SMM and S5D using REF as the reference ensemble forecast. This way, red (resp. blue) areas show where SMM and S5D have higher (resp. lower) skill than REF. No clear pattern emerges regarding skill improvements over the North Atlantic, although oftentimes SMM and S5D do improve model skill. One must bear in mind that skill is quite limited to begin with in the REF ensemble, as reminded in the manuscript.

3) Technical corrections:

Thank you for pointing out some errors left in the manuscript. Regarding the statistical significance, figures have been redone using larger stippling to highlight better the areas where differences/results are significant.

Regarding the reference p3, l21: the author's last name is "Salas y Melia", this isn't a typo.

4) Changes to the manuscript:

Changes to the manuscript can be tracked in the supplement to this comment, with red crossed text indicating suppressions and underlined blue text indicating additions with respect to the original submission.

Please also note the supplement to this comment:
http://www.geosci-model-dev-discuss.net/gmd-2015-270/gmd-2015-270-AC1-supplement.pdf

[Figure]

**Fig. 1.** Relative Z500 absolute bias for DJF 1979-2012 re-forecasts SMM (left) and S5D (right) with respect to REF. Blue areas show where bias is reduced regardless of sign.

---

## Author Comment (AC2) · 2 May 2016

Reply to interactive comment by anonymous referee #2 by Lauriane Batté

We wish to start by thanking the reviewer for his/her constructive comments on our manuscript.

1) Reply to comments:

"I commend the authors for being clear about the limitations of their technique and not overselling their results. Yet, I think the manuscript would benefit from establishing clear

expectations of the technique – and the abstract, the introduction and the conclusions are not very coherent. [. . .] I suggest that the authors a. formulate a coherent goal for the manuscript b. include [. . .] a discussion where they see the further potential of the technique."

Thank you for this valuable comment. We tried to address these points by reformulating the abstract, introduction and conclusion.

"Also, as a comment, I think the differences/improvements in figure 6 are not small."

I agree. This aspect of model improvement with the introduction of these perturbations is primordial, also in the sense that it seems to translate into improvements in the representation of North Atlantic weather regimes.

"The split up of the model (experiment) description between section 2.1 and 4.1 was not entirely intuitive to me. Could the two sections be combined within section 2? Also, is the horizontal resolution mentioned anywhere?

I originally placed the experiment description in 4.1, since some settings for the per-turbation frequency were motivated by analyses presented in section 3. To take into account your comment, I combined the experiment description in section 2.1 (regard-ing initial conditions, re-forecast period, ensemble size) with the details on stochastic dynamics settings presented originally in 4.1 into a section 2.3 on seasonal re-forecast experiments description.

Thank you for pointing out that the horizontal resolution isn't mentioned. Section 2.1 was corrected accordingly.

"I think section 4.5.1 could do with a mentioning of the recent results of NAO skill (e.g. Scaife et al., Butler et al.; including Weisheimer et al., if the authors wish to question the results)."

I included these references and additional discussion in section 4.5.1.

"I noticed that the references to figures are sometimes with "Fig." and sometimes with "figure". Also, are the supplementary figures cited (in the right order)? Maybe I overlooked it, but where is figure S2 cited?"

My intention was to use "Fig." when inside a sentence, and "Figure" at the beginning of the sentence. Some occurrences may have been left out, but as you mentioned, this should undergo proofreading later on. Figure S2 (now S3) is cited in page 9 – line 15 alongside figure S1 (now S2). The formulation wasn't very clear, I clarified this in the revised version of the manuscript by "Results for 500 hPa geopotential height are shown in supplementary fig. S2 for November and fig. S3 for DJF."

2) Changes in the manuscript

Changes to the manuscript can be tracked in the supplement to this comment, with red crossed text indicating suppressions and underlined blue text indicating additions with respect to the original submission.

Please also note the supplement to this comment:
http://www.geosci-model-dev-discuss.net/gmd-2015-270/gmd-2015-270-AC2-supplement.pdf

**Supplement:**

[revised manuscript text omitted]

---

## Author Comment (AC4) · 4 May 2016

Regarding the variables nudged, the answer provided in my original response was incorrect. Sorry about this. The rotational component is perturbed, and divergence part is not perturbed. The nudged variable is indeed vorticity, using the same tau=1 month as for the other variables. The text should be corrected as suggested by reviewer 3. Regards, Lauriane BATTE

---

## Author Response (AR1)

**Reviewer comments and point-by-point response**
**Lauriane Batté**

The following author's response is structured as follows : part I) lists the different reviewer comments and response (author comments uploaded in the interactive discussion), part II) lists the changes made to the manuscript, and part III) is the manuscript in track changes mode (produced with latexdiff) so as to highlight these changes in the text.

**I) Point-by-point response to reviewer comments**

**1) Reviewer 1 comments**

**a) General comments**

*"Unfortunately, the impact of the stochastic dynamic technique is small. I suggest to expand the discussion in the conclusions, why the impact is small and why the results of forcing with monthly mean tendencies is so similar to using 5d-consecutive tendencies."*

Our hypothesis, based on this study and previous work on the technique, is that the main impact of our perturbations does derive from the systematic error corrections encompassed in the perturbation term. This is why on average, 5d-consecutive tendencies have the same effect on seasonal forecast quality than the monthly mean tendencies.

Regarding the limited impact in both setups on seasonal forecasting skill, this is most probably related to the weak constraint in our preliminary experiment. With a previous version of the model, other settings for the nudged preliminary run were tested, using a stronger constraint. However, our feeling was that since we were nudging towards ERA-Interim, using too strong a nudging could be a drawback, in the sense that we would be drawing the model away from its own equilibrium (and more towards that of the ECMWF model), and the terms would be less representative of long-term model errors. Were we to have a reanalysis based on the ARPEGE-Climate model, we could consider using stronger nudging and explore the impact this has then when applying the corresponding perturbations in seasonal forecast mode.

**b) Reply to specific comments**

*"Move discussion on page 13, l13-15 to conclusions and expand. Is there a pattern that SMM and S5D have similar impact on mean statistics, but S5D a larger impact on statistics involving the second moment?"*

I have rearranged the conclusions to take into account this comment. Regarding the impact on statistics involving the second moment, our results with respect to weather regime duration, etc. suggest that differences are also small between S5D and SMM. This could be due to the fact that our nudging is quite weak and is a perspective for future work.

*"It would be interesting to see a map of a particular 5D-tendency to get a feeling for the spatial correlation scales."*

I included this in the supplementary information of the article as (new) figure S1, and commented this in the article (section 3.3).

*"It might be helpful to plot the differences SMM-REF and S5D-REF for figures 5, 6 and 10 to see if*

*there is a coherent regional signal. As the manuscript admits, the absolute plots look very similar."*

Figure 5 shows the relative absolute bias of SMM and S5D with respect to that of REF in the middle and bottom rows (meaning that blue areas show where bias is reduced, and red areas where bias is enhanced, regardless of sign). Over the Northern Hemisphere extra-tropics, the main impression is that SST bias is reduced with our technique, whereas results are more contrasted for precipitation (patchy areas, general reduction of bias over the mid-latitudes, and increase in precipitation bias over the Arctic).

[Figure]

*Fig C1: Relative Z500 absolute bias for DJF 1979-2012 re-forecasts SMM (left) and S5D (right) with respect to REF. Blue areas show where bias is reduced regardless of sign.*

In my opinion, the different figures in figure 6 are not that similar, they show a substantial reduction of the Z500 bias over most of the Northern Hemisphere extra-tropics. As additional information, figure C1 included in this comment shows the relative absolute bias for DJF Z500 over the re-forecast period for experiments SMM and S5D with respect to experiment REF. Similar information is available for the reader in the supplementary figure S2. I clarified the sentence referring to these results and figure S2 in the revised version of the manuscript.

Figure 10 now shows the CRPSS for REF with respect to reference data climatological probabilities, and CRPSS for SMM and S5D using REF as the reference ensemble forecast. This way, red (resp. blue) areas show where SMM and S5D have higher (resp. lower) skill than REF. No clear pattern emerges regarding skill improvements over the North Atlantic, although oftentimes SMM and S5D do improve model skill. One must bear in mind that skill is quite limited to begin with in the REF ensemble, as reminded in the manuscript.

**c) Technical corrections**

Thank you for pointing out some errors left in the manuscript. Regarding the statistical significance, figures have been redone using larger stippling to highlight better the areas where differences/results are significant.
Regarding the reference p3, l21: the author's last name is "Salas y Melia", this isn't a typo.

**2) Reviewer 2 comments**

*"I commend the authors for being clear about the limitations of their technique and not overselling their results. Yet, I think the manuscript would benefit from establishing clear expectations of the technique – and the abstract, the introduction and the conclusions are not very coherent. […] I*

*suggest that the authors a. formulate a coherent goal for the manuscript b. include […] a discussion where they see the further potential of the technique."*

Thank you for this valuable comment. We tried to address these points by reformulating the abstract, introduction and conclusion.

*"Also, as a comment, I think the differences/improvements in figure 6 are not small."*

I agree. This aspect of model improvement with the introduction of these perturbations is primordial, also in the sense that it seems to translate into improvements in the representation of North Atlantic weather regimes.

*"The split up of the model (experiment) description between section 2.1 and 4.1 was not entirely intuitive to me. Could the two sections be combined within section 2? Also, is the horizontal resolution mentioned anywhere?*

I originally placed the experiment description in 4.1, since some settings for the perturbation frequency were motivated by analyses presented in section 3. To take into account your comment, I combined the experiment description in section 2.1 (regarding initial conditions, re-forecast period, ensemble size) with the details on stochastic dynamics settings presented originally in 4.1 into a section 2.3 on seasonal re-forecast experiments description.

Thank you for pointing out that the horizontal resolution isn't mentioned. Section 2.1 was corrected accordingly.

*"I think section 4.5.1 could do with a mentioning of the recent results of NAO skill (e.g. Scaife et al., Butler et al.; including Weisheimer et al., if the authors wish to question the results)."*

I included these references and additional discussion in section 4.5.1.

*"I noticed that the references to figures are sometimes with "Fig." and sometimes with "figure". Also, are the supplementary figures cited (in the right order)? Maybe I overlooked it, but where is figure S2 cited?"*

My intention was to use "Fig." when inside a sentence, and "Figure" at the beginning of the sentence. Some occurrences may have been left out, but as you mentioned, this should undergo proofreading later on. Figure S2 (now S3) is cited in page 9 – line 15 alongside figure S1 (now S2). The formulation wasn't very clear, I clarified this in the revised version of the manuscript by "Results for 500 hPa geopotential height are shown in supplementary fig. S2 for November and fig. S3 for DJF."

**3) Reviewer 3 comments**

**a) Reply to general comments**

*"I believe there is a general problem with the use of the "initial" when τ is as long as 30 days. With such a weak nudging this term can not be said to represent initial tendency errors but rather long term secondary adjustments (that luckily seem to have some positive impact). This is of cause because, on a monthly time scale, initial forcing in terms of e.g. potential vorticicy will show up far away via Rossby wave dispersion. As an example consider the right column of Figure 2: These corrections could very well be due to "real" initial errors in the tropics. It is therefore suggested not to use the expression "initial" tendency errors. One could, e.g., call it model drift error."*

This is a very interesting comment. The use of "initial" in our tendency error estimations originates from the previous version of our method, which used a much stronger nudging. You are right that with a 30 day nudging strength, the differences will be more representative of longer term errors. We accepted and included the formulation you suggested (model drift error).

*b) Reply to specific comments*

We include the minor corrections you suggested to our manuscript. More details are included below where appropriate.

*"Page 4, line 4 : I presume you mean "not to perturb the divergent component" instead of "not to perturb the rotational component" (since vorticity represents the rotational part)."*

Actually, line 3 of page 4 should read « streamfunction » instead of vorticity! Sorry for the confusion and thank you for pointing this out. We corrected accordingly.

*"Page 5 ff: Probably not only the magnitude but also the shape of the spectra are quite dependent on τ . A short discussion on this would be relevant."*

You're right. We included a few lines on this in section 3.1.

*"Page 8, Section 4.2: It would be relevant to show - or at least discuss - the bias in the initial nudged simulations as well. Ideally the mean error of these runs should be small. But with the large value of τ one would suspect that this is not the case."*

A detailed discussion on the impact of the strength of the nudging on the quality of the nudged re-forecast runs is somewhat beyond the scope of this manuscript, in our opinion. Based on past work with the method using a stronger nudging, the nudged simulations have (by construction) a much closer mean climate to that of the reanalysis dataset used as a reference, however since this reanalysis is not based on the same atmospheric model as our forecasting system, the model error estimates are not truly representative of errors in forecast mode. Using settings from the previous version of the method described in Batté and Déqué 2012, we found some adverse effects on ENSO prediction skill with our new coupled system. This motivates the use of much "looser" nudging to let the model drift away from reference data; however it is true that the bias (and skill estimates, although with only one member) are degraded with this setting. Note however that the bias and skill of the nudged run is (in most cases) significantly improved with respect to our REF experiment. We have yet to test an intermediate solution as a trade-off between both nudging strengths (that discussed in this paper and in the 2012 GRL), to see the impact on forecast quality.

We included a sentence relative to the bias of the nudged reforecast in section 4.1.

*"Page 11, line 32: "... not capture its interannual variability". One would guess that it could also be large if the model has a bias. Any bias could be subtracted before calculating RMSE. This would probably give considerably smaller RMSE's.*
*Page 13, lines 17-21: Also here it could be relevant to eliminate the impact of bias."*

This is done in our computation of the RMSE. I clarified this where the RMSE score is discussed.

*"Page 14, line 3: You could provide a quantitative estimate of the uncertainties in the correlations!"*

Based on bootstrapping over the years of the re-forecast period, the 95% significance intervals (with 10000 draws) for the NAO correlation are [0.119, 0.641] for REF, [0.009, 0.656] for SMM and [0.181, 0.797] for S5D. The interval is wider in the case of SMM and slightly shifted towards higher values in the case of S5D, however given the broad intervals in this case it seems difficult to draw any firm conclusion.

*"Page 14, lines 22-23: Why is there no SMM in Table 2 (and 3)?"*

I have fixed this in the revised manuscript. As you will see, results are very similar between both versions S5D and SMM tested.

*"Page 15, Section 4.5.3: I think this section can be removed. It does not add much to the findings already described."*

We feel examining weather regime frequency prediction skill (or lack thereof) is the next logical step to assessing the impact of the perturbations on North Atlantic large-scale variability. Although results are very limited, this is why we chose to include these in the paper. We think the paragraph should be kept in the manuscript.

**II) Changes in the manuscript**

**Abstract**
The abstract was re-written following reviewer 2's concern on the coherence between the abstract, introduction and conclusion, mainly by re-arranging the second and third paragraph of the abstract.

**Introduction**
The term "initial tendency" was rephrased following a suggestion from reviewer 3. (This was also done throughout the manuscript).
Parts of the introduction (on the aim of the paper) were rewritten following concerns from reviewer 2.

**Section 2**
*Section 2.1* was enhanced with information on the model horizontal resolution in the atmosphere and ocean components, and presentation of the seasonal re-forecasting framework was moved to a new *section 2.3*.

**Section 3**
Additional comments on the dependence on tau of the shape of the spectra were added in *section 3.1* following reviewer 3 comments.
A sentence pointing to 5-day consecutive tendencies plotted in the supplementary material was included in *section 3.3*.

**Section 4**
*Section 4.1* was removed and merged with parts of section 2.1 into a separate *section 2.3* (see above).
The bias of the nudged re-forecast was briefly discussed in section 4.1 (formerly 4.2).
Equation (8) was moved to where it is referred to in the manuscript.
Since figure 10 was modified, its description was modified accordingly in section 4.3 (formerly 4.4).
A comment in section 4.3 was moved to conclusions following remarks by reviewer 1.
Additional information and references on NAO seasonal prediction skill were included in section 4.4.1.
Results for weather regime statistics and prediction skill with the SMM ensemble were included in the paper, and text for sections 4.4.2 and 4.4.3 were enhanced accordingly following a question from reviewer 3.

**Conclusions**

Conclusions were expanded on the very little differences found between experiments SMM and S5D presented in the paper.

**References**

Additional references were included on NAO seasonal forecast skill: Doblas-Reyes et al. 2003, Butler et al. 2016, Scaife et al. 2014, Riddle et al. 2013, Stockdale et al. 2015.

**Tables**

*Tables 2 and 3* were modified to include results for the SMM ensemble.

Caption for *figure 5* was corrected.

*Figures 7 and 9* were modified to increase visibility of the statistically significant results.

*Figure 10* was modified to show CRPSS using the REF ensemble as a reference in the case of SMM and S5D experiments, the caption was modified accordingly.

*Figure 13* was modified to include results for SMM, and the caption changed to account for this modification.

**Supplementary material**

An additional figure was included in the supplementary material for this manuscript, and the numbering of the supplementary figures changed accordingly in the text.

**III) Track changes**

Please refer to the supplementary information provided in the author comments to each reviewer.

[revised manuscript text omitted]

(a) TAS REF vs. clim      (b) Precipitation REF vs. clim      (c) Z500 REF vs. clim

(d) TAS SMM vs. REF      (e) Precipitation SMM vs. REF      (f) Z500 SMM vs. REF

(g) TAS S5D vs. REF      (h) Precipitation S5D vs. REF      (i) Z500 S5D vs. REF

**Figure 10.** (a-c) DJF continuous ranked probability skill score (CRPSS) for the REF , SMM and S5D experiments (top to bottom rows, respectively) experiment near-surface air temperature, precipitation and Z500(from left to right). Areas in red/blue indicate where the model skill is higher/lower than a reference forecast using climatology. (d-f, g-i) Same as (a-c) but for SMM and S5D experiments (middle and bottom rows, respectively) computing CRPSS with REF as a reference.

[Figure]

**Figure 11.** Evolution of spread (dots) and RMSE (lines) with forecast time for 500 hPa geopotential height over Europe in experiments REF (red), SMM (blue) and S5D (green).

[Figure]

**Figure 12.** DJF NAO index computed with ERA-Interim 500 hPa geopotential height (black lines) and boxplots of ensemble re-forecasts REF (gray), SMM (blue) and S5D (red) NAO indices computed by projecting model anomalies on the ERA-Interim NAO pattern. Anomalies and NAO indices are computed in cross-validation mode. The correlation between the ensemble mean and ERA-Interim index is shown in the top left corner of the figure.

[Figure]

**Figure 13.** Frequency of weather regime transitions (in %) computed by discarding regimes shorter than 3 days (considered as transition days) over DJF 1979–2012. Results are shown for ERA-Interim reanalysis (in black) and experiments REF, SMM and S5D for DJF 1979–2012(in grey, blue and red, respectively).